# Coupling carbon and energy fluxes in the North Pacific Subtropical Gyre

Eric Grabowski [1,2], Ricardo M. Letelier [1,3], Edward A. Laws[1,4] & David M. Karl [1,2]

The major biogeochemical cycles of marine ecosystems are driven by solar energy. Energy that is initially captured through photosynthesis is transformed and transported to great ocean depths via complex, yet poorly understood, energy flow networks. Herein we show that the chemical composition and specific energy (Joules per unit mass or organic carbon) of sinking particulate matter collected in the North Pacific Subtropical Gyre reveal dramatic changes in the upper 500 m of the water column as particles sink and age. In contrast to these upper water column processes, particles reaching the deep sea (4000 m) are energy-replete with organic carbon-specific energy values similar to surface phytoplankton. These enigmatic results suggest that the particles collected in the abyssal zone must be transported by rapid sinking processes. These fast-sinking particles control the pace of deep-sea benthic communities that live a feast-or-famine existence in an otherwise energy-depleted habitat.

[1] Daniel K. Inouye Center for Microbial Oceanography: Research and Education (C-MORE), University of Hawaii at Manoa, Honolulu 96822 HI, USA. [2] School of Ocean and Earth Science and Technology, University of Hawaii at Manoa, Honolulu 96822 HI, USA. [3] College of Earth, Ocean and Atmospheric Sciences, Oregon State University, Corvallis 97331 OR, USA. [4] College of the Coast & Environment, Louisiana State University, Baton Rouge 70803 LA, USA. Correspondence and requests for materials should be addressed to D.M.K. (email: dkarl@hawaii.edu)

The ocean's biological carbon pump (BCP) is an integral component of the global carbon (C) cycle, largely responsible for the long-term sequestration of carbon dioxide ($CO_2$) into the mesopelagic and abyssal zones[1]. Because nearly all biological and biogeochemical processes in the sea, including the BCP, are solar-powered with light energy initially transformed into chemical potential energy via the process of photosynthesis, there is an inextricable, yet poorly characterized, linkage between C fluxes and energy fluxes in the sea[2]. In marine ecosystems, pigments (e.g., chlorophylls, carotenoids, and biliproteins) contained within phototrophic prokaryotic and eukaryotic microorganisms absorb solar energy through a complex series of light-dependent reactions, and transfer electrons ($e^-$) from donor to acceptor molecules. During this process, water ($H_2O$) is the $e^-$ donor being oxidized to molecular oxygen ($O_2$) in stoichiometric proportion to photon capture (i.e., $2H_2O \rightarrow 4H^+ + 4e^- + O_2$). In a second, light-independent series of reactions, carbon dioxide ($CO_2$) is reduced to carbohydrate[3], a form of potential energy that is ultimately used to build other biomolecules and to sustain heterotrophic processes throughout the water column and into the abyssal sediments. In marine ecological studies, photosynthesis is typically estimated by measuring $O_2$ production or $CO_2$ reduction per unit volume and per unit time during incubation experiments[4,5]. However, these $O_2$ or $CO_2$ fluxes alone are insufficient to track energy within the surface ocean, or to quantify potential energy export from the sunlit zone of maximum production into the mesopelagic and abyssal zones in the form of sinking particulate matter (SPM).

We have recently estimated gross primary production using $^{18}O$-$H_2O$ (refs. [4,5]) and the total photosynthetic pigment absorption of solar energy[6] in the surface waters at Station ALOHA in the North Pacific Subtropical Gyre (NPSG)[7]. The calculated quantum yield of ~0.1 mol $O_2$ evolved per mol quanta absorbed by photosynthetic pigments[6] is in excellent agreement with laboratory-derived values[8], and equates to a mean photosynthetic energy capture of ~33 kJ m$^{-2}$ d$^{-1}$ (see Methods). In September 2013, we deployed a free-drifting array of sediment traps to collect SPM over a range of 12 depths (100–500 m) for a period of 9.1 days. Particle mass, C (organic [OC], inorganic [IC], and black [BC]), hydrogen (H), nitrogen (N), phosphorus (P) as well as total potential energy were used to calculate downward fluxes and to determine changes as particles sink from the sunlit region of the water column and are remineralized during transit through the mesopelagic zone. Potential energy was measured as heat released upon combustion using an oxygen bomb calorimeter. In 2016, we deployed a bottom-moored, sequencing sediment trap to collect SPM at 4000 m (800 m above the seabed) during both winter and summer seasons. In addition, we also sampled the Station ALOHA sediment trap archive to measure the chemical composition and total potential energy of deep-sea SPM from winter and summer seasons in 1998 and 2000 (ref. [9]).

Measurements of heats of combustion (enthalpy, measured in Joules) should provide upper constraints of total potential energy in the SPM collected at various depths in the water column. Although the Gibbs energy of combustion that is available to microorganisms is expected to be less than the measured enthalpy, a review of relevant literature on the thermodynamics of aerobic microbial growth concluded that enthalpy changes are generally close to the Gibbs energy changes[10]. Therefore, measurements of the enthalpy of SPM can provide reasonable estimates of the total energy (measured in Joules), specific energy (J mg$^{-1}$ mass or J mg$^{-1}$ OC) and downward energy transport (J m$^{-2}$ d$^{-1}$) associated with SPM collected at different depths in the water column.

We present several lines of evidence showing that much of the energy is lost in the upper mesopelagic zone via selective remineralization as particles sink and age, leaving behind an energy-depleted organic matter pool containing black carbon. The SPM collected at the 100 and 500 m reference depths contain ~6.5% and ~2.6%, respectively, of the total energy that is produced through photosynthesis in the overlying water column. The chemical composition and specific energy content of the SPM at 500 m is consistent with highly oxidized, low-potential energy organic matter. While the flux of total energy to the abyss (4000 m reference depth) is very low (<0.5% of total energy captured via photosynthesis), the OC-specific energy for abyssal particles is on par with near-surface energy-replete particles. These observations support the existence of two fundamentally different classes of sinking particles, namely energy-deplete, slow-sinking particles and energy-replete, fast-sinking particles. These BCP energy flux data, connecting the euphotic zone to mesopelagic and abyssal habitats, are the first of their kind for any marine environment.

## Results

**Elemental fluxes in the upper ocean.** The fluxes of SPM mass and C were highest at 100 m and decreased with increasing water depth (Fig. 1 and Table 1), in accordance with previous field observations at or near Station ALOHA[11–13]. The flux profiles were fit to a normalized power function of the form, $F_z = F_{100} (z/100)^b$, where $z$ is water depth (m) and $F$ and $F_{100}$ are fluxes at reference depths $z_m$ and 100 m, respectively (ref. [11]; see Supplementary Fig. 1 for individual plots). The total C (TC) fraction was comprised of three separate components: OC, the major fraction of which decreased significantly (~75%) over the 100–500 m depth range, and two less abundant (<5% by mass at the 100-m reference depth) C components (IC and BC), which both increase with depth as percentages of TC (Fig. 1 and Table 1). IC (i.e., calcium carbonate) is produced by a variety of marine organisms and is expected to be present in SPM, whereas BC is most likely derived from allochthonous sources as a by-product of incomplete combustion of organic matter[14,15]. While BC is technically part of the OC pool, it has a pyrogenic origin and is more resistant to thermal and chemical degradation than the autochthonous OC fraction (see Methods). The elemental composition of SPM also changed systematically with increasing depth, becoming more OC-enriched relative to N and P (Table 1 and Supplementary Fig. 2). A comparison of the elemental composition of SPM at 100 m to that at 500 m indicates that the loss of mass from sinking particles to the water column over this depth range had a mean molar stoichiometry of $OC_{162}H_{271}N_{28}P_1$, despite the changing stoichiometry of the residual SPM (Table 1). This indicates a selective remineralization of SPM, presumably by aerobic microbial decomposition, as particles sink and age.

**Energy fluxes in the upper ocean.** The downward flux of energy associated with SPM, expressed as J m$^{-2}$ d$^{-1}$, also displayed a significant decrease with depth (Fig. 2 and Table 2). The downward energy flux measured at 100 m is equivalent to ~6.5% of the solar energy captured via photosynthesis at Station ALOHA (i.e., 2138 J m$^{-2}$ d$^{-1}$ compared to 33,054 J m$^{-2}$ d$^{-1}$; Figs. 2 and 3, and Table 2). These data provide direct evidence for an efficient energy-dissipative, remineralization-intensive euphotic zone (0–100 m) where ~95% of the solar energy captured and stored during the process of photosynthesis is locally transformed, assuming steady-state conditions. The energy contained within the SPM that does escape the upper euphotic zone exhibits a significant depth-dependent decrease to a downward flux of ~125 J m$^{-2}$ d$^{-1}$ at 500 m (Fig. 2 and Table 2). This observed loss within the 100–500 m region of the water column is equivalent to ~95% of the energy that is exported from the euphotic zone via SPM (Table 2).

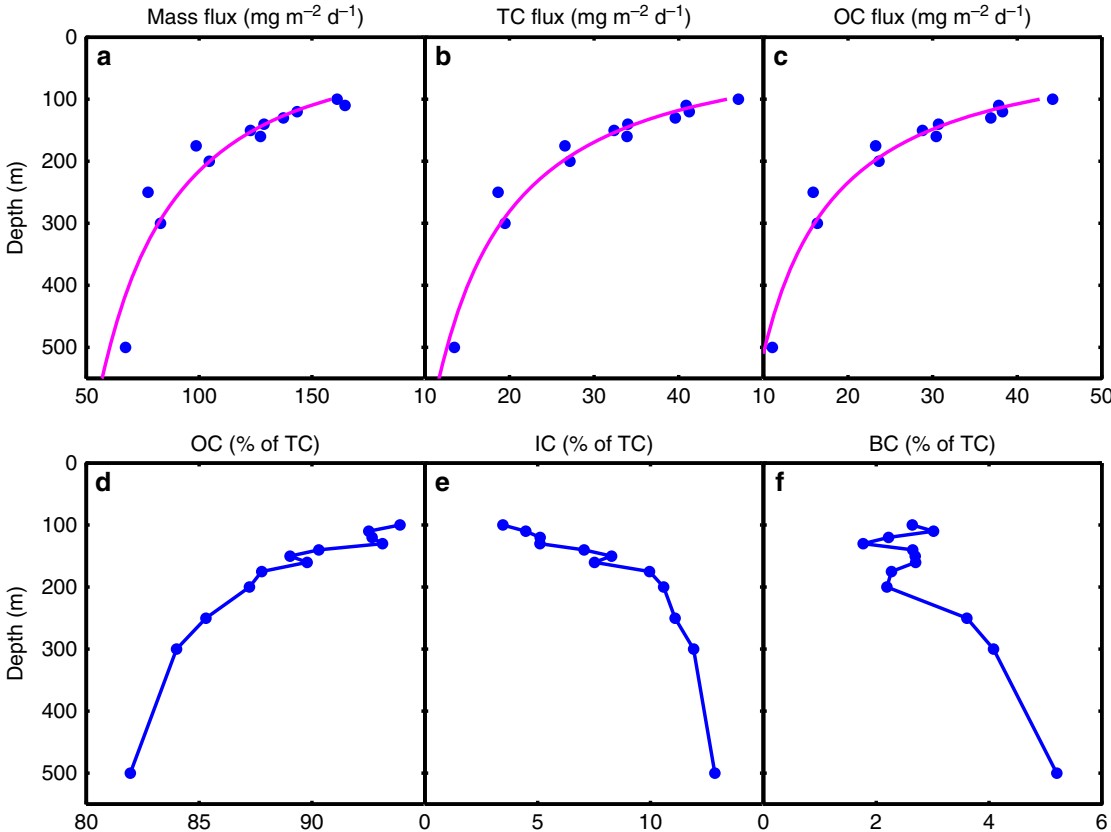

**Fig. 1** Mass and carbon fluxes. Flux profiles of **a** mass, **b** total carbon (TC), and **c** organic carbon (OC). The magenta curves are the best fits to a log–log transformed normalized power function of the form $F_z = F_{100}(z/100)^b$ where $F_z$ is flux at depth $z$(m), $F_{100}$ is flux at 100 m and $b$ is the coefficient of flux attenuation (ref. [11]; Supplementary Fig. 1). The best fit parameters are: mass flux, $F_{100} = 158.5$ mg m$^{-2}$ d$^{-1}$, $b = -0.60$, $r^2 = 0.85$; TC flux, $F_{100} = 45.71$ mg C m$^{-2}$ d$^{-1}$, $b = -0.80$, $r^2 = 0.93$; OC flux, $F_{100} = 42.66$ mg C m$^{-2}$ d$^{-1}$, $b = -0.89$, $r^2 = 0.93$. Vertical fluxes of **d** OC, **e** IC and **f** BC expressed as percentages of TC for each depth

| **Table 1 Total mass and elemental fluxes of sinking particulate matter** | | | | | | | |
|---|---|---|---|---|---|---|---|
| Sediment trap depth (m) | Mass flux[a] (mg m$^{-2}$ d$^{-1}$) | Carbon flux (mg m$^{-2}$ d$^{-1}$) | | | Hydrogen flux (mg m$^{-2}$ d$^{-1}$) | Nitrogen flux (mg m$^{-2}$ d$^{-1}$) | Phosphorus flux (mg m$^{-2}$ d$^{-1}$) | OC:H:N:P (molar) |
| | | OC | IC | BC | | | | |
| 100 | 161.2 ± 13.0 | 44.2 ± 0.9 | 1.62 ± 0.00 | 1.24 ± 0.03 | 6.33 ± 0.11 | 8.18 ± 0.21 | 0.62 ± 0.01 | 185:317:29:1 |
| 110 | 164.6 ± 13.1 | 37.8 ± 1.0 | 1.83 ± 0.10 | 1.23 ± 0.08 | 5.61 ± 0.11 | 7.15 ± 0.13 | 0.48 ± 0.01 | 204:362:33:1 |
| 120 | 143.4 ± 15.2 | 38.3 ± 1.0 | 2.11 ± 0.04 | 0.87 ± 0.08 | 5.49 ± 0.20 | 6.93 ± 0.30 | 0.44 ± 0.02 | 223:387:35:1 |
| 130 | 137.3 ± 9.6 | 36.9 ± 0.8 | 2.02 ± 0.11 | 0.69 ± 0.04 | 5.26 ± 0.12 | 6.42 ± 0.16 | 0.43 ± 0.01 | 223:379:33:1 |
| 140 | 128.8 ± 14.5 | 30.7 ± 0.3 | 2.40 ± 0.17 | 0.89 ± 0.06 | 4.48 ± 0.06 | 5.36 ± 0.03 | 0.35 ± 0.01 | 229:397:34:1 |
| 150 | 122.8 ± 7.9 | 28.8 ± 1.1 | 2.68 ± 0.15 | 0.86 ± 0.04 | 4.16 ± 0.15 | 4.95 ± 0.20 | 0.32 ± 0.02 | 230:403:34:1 |
| 160 | 127.2 ± 10.2 | 30.4 ± 0.6 | 2.55 ± 0.16 | 0.92 ± 0.09 | 4.45 ± 0.01 | 4.98 ± 0.09 | 0.31 ± 0.02 | 254:445:36:1 |
| 175 | 98.7 ± 7.7 | 23.3 ± 0.6 | 2.65 ± 0.09 | 0.60 ± 0.01 | 3.37 ± 0.11 | 3.60 ± 0.11 | 0.21 ± 0.01 | 290:497:39:1 |
| 200 | 104.5 ± 6.3 | 23.7 ± 0.4 | 2.88 ± 0.08 | 0.59 ± 0.04 | 3.39 ± 0.07 | 3.84 ± 0.07 | 0.23 ± 0.00 | 272:457:38:1 |
| 250 | 77.4 ± 5.9 | 15.9 ± 0.4 | 2.07 ± 0.07 | 0.68 ± 0.01 | 2.35 ± 0.05 | 2.52 ± 0.06 | 0.14 ± 0.01 | 289:520:39:1 |
| 300 | 82.8 ± 2.5 | 16.4 ± 0.8 | 2.32 ± 0.05 | 0.79 ± 0.01 | 2.45 ± 0.10 | 2.38 ± 0.12 | 0.13 ± 0.01 | 337:584:42:1 |
| 500 | 67.5 ± 6.4 | 11.1 ± 0.1 | 1.73 ± 0.05 | 0.70 ± 0.02 | 1.69 ± 0.01 | 1.50 ± 0.04 | 0.09 ± 0.01 | 319:582:37:1 |

[a]All values are reported as mean ±1 standard deviation

Both the mass- and OC-specific energy of the residual SPM decrease significantly with depth between 100 and 500 m. This systematic trend toward an increasingly more energy-depleted residual SPM is consistent with the changes in bulk elemental stoichiometry and the higher BC content with depth. However, there are important distinctions between the depth-dependent changes in mass fluxes versus changes in SPM specific energy. Whereas the C:N and C:P ratios change continuously throughout the flux profile, the specific energy of the residual SPM exhibits a more abrupt change below 250 m, especially for the OC-specific energy (Figs. 2 and 3, and Table 2). SPM collected above the compensation irradiance, the depth where net photoautotrophic solar energy capture is zero over a 24-h period (equal to a solar energy flux of 0.054 mol photons m$^{-2}$ d$^{-1}$ or a depth of approximately 175 m; ref. [16]) has a nearly constant mass- and OC-specific energy with mean values of 11.83 (s.d. = 1.28, $n = 8$) J mg$^{-1}$ and 47.53 (s.d. = 2.18, $n = 8$) J mg$^{-1}$, respectively (Fig. 2). However, below the compensation irradiance (>175 m), there is a

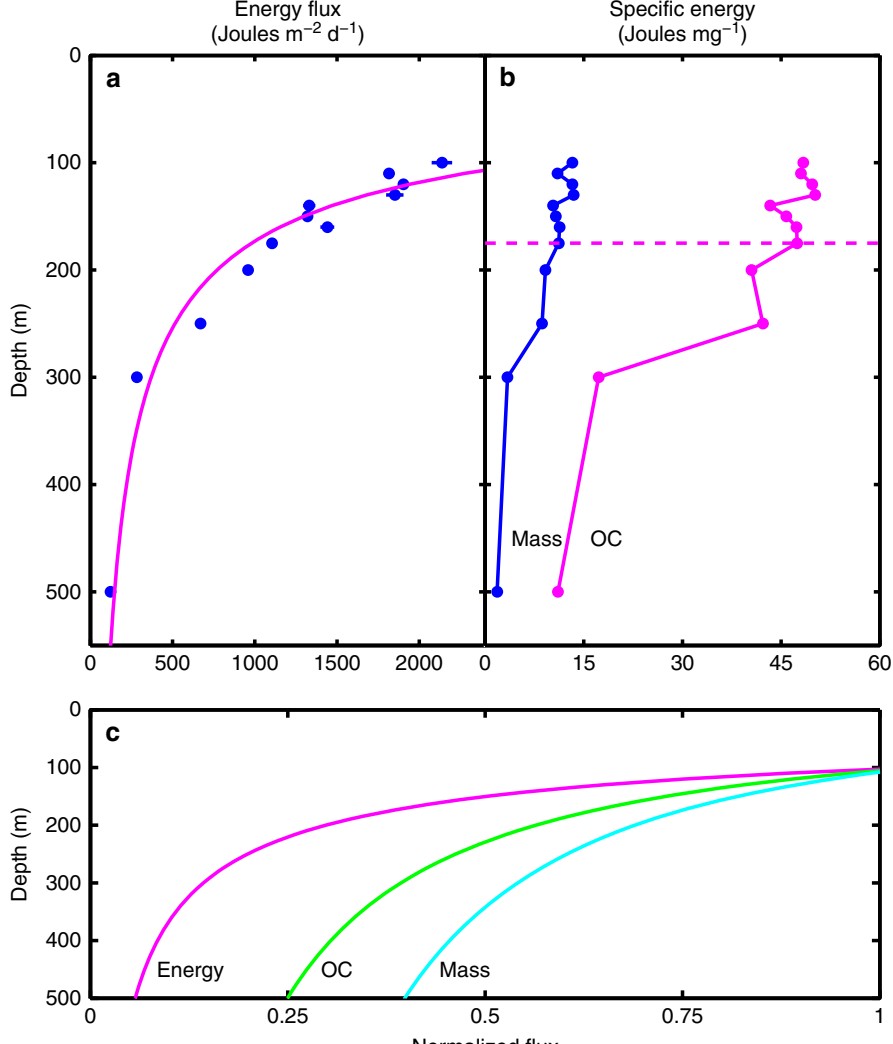

**Fig. 2** Energy fluxes. **a** Energy flux and **b** specific energy of SPM expressed in mass-specific (blue symbols) and OC-specific (magenta symbols) units. The magenta curve on the left is the best fit for the equation presented in Fig. 1 (also see Supplementary Fig. 2). The best fit parameters are: $F_{100} = 2701.6$ $J\,m^{-2}\,d^{-1}$, $b = -1.81$, $r^2 = 0.90$. The horizontal magenta dashed line on the right is the approximate depth of the photosynthetic compensation irradiance (see text for details). **c** Schematic representation of the flux profiles and flux attenuation coefficients for mass (blue curve, $b = -0.60$), OC (green curve, $b = -0.89$), and energy (magenta curve, $b = -1.81$).

**Table 2 Total energy fluxes and specific energy of sinking particulate matter**

| Depth (m) | Energy flux[a] ($J\,m^{-2}\,d^{-1}$) | Specific energy | |
|---|---|---|---|
| | | ($J\,mg^{-1}$ mass) | ($J\,mg^{-1}$ OC) |
| 100 | 2137.4 ± 62.3 | 13.26 | 48.37 |
| 110 | 1816.2 ± 32.3 | 11.03 | 48.03 |
| 120 | 1902.8 ± 30.0 | 13.27 | 49.72 |
| 130 | 1851.3 ± 52.7 | 13.48 | 50.19 |
| 140 | 1331.2 ± 37.2 | 10.34 | 43.37 |
| 150 | 1319.2 ± 25.7 | 10.75 | 45.79 |
| 160 | 1440.8 ± 42.1 | 11.32 | 47.35 |
| 175 | 1105.0 ± 27.4 | 11.20 | 47.44 |
| 200 | 959.7 ± 21.1 | 9.18 | 40.50 |
| 250 | 671.2 ± 4.7 | 8.67 | 42.21 |
| 300 | 282.5 ± 3.8 | 3.41 | 17.27 |
| 500 | 122.6 ± 6.7 | 1.82 | 11.09 |

[a]All values are reported as mean ±1 standard deviation

significant and systematic decrease in the mass- and OC-specific energy of the residual SPM to much lower values. This pattern may reflect a selective light- and depth-dependent utilization of energy-replete organic compounds from the SPM pool as particles age. Based on these observations, we estimate that the SPM remineralized in the 100–500 m region of the water column has an average specific energy of ~60 $J\,mg^{-1}$ OC. This value is slightly higher than the mean value for SPM collected at 100 m (Table 2), but is nearly identical to the OC-specific energy of carbon fixed in photosynthesis observed during the peak of a natural phytoplankton bloom[17]. These results, along with the observed changes in the elemental composition, suggest that much of the freshly produced SPM is selectively and rapidly remineralized as particles age while sinking. Although the molecular composition of SPM in the mesopelagic zone is largely unknown, the OC-specific energy at 500 m, 11.09 $J\,mg^{-1}$ (Table 2), is indicative of an extremely energy-depleted residual SPM compared to near-surface values. In addition, the percentage increase in the contribution of BC to total SPM carbon with depth, from ~2% of TC in the euphotic zone to >5% at the 500 m

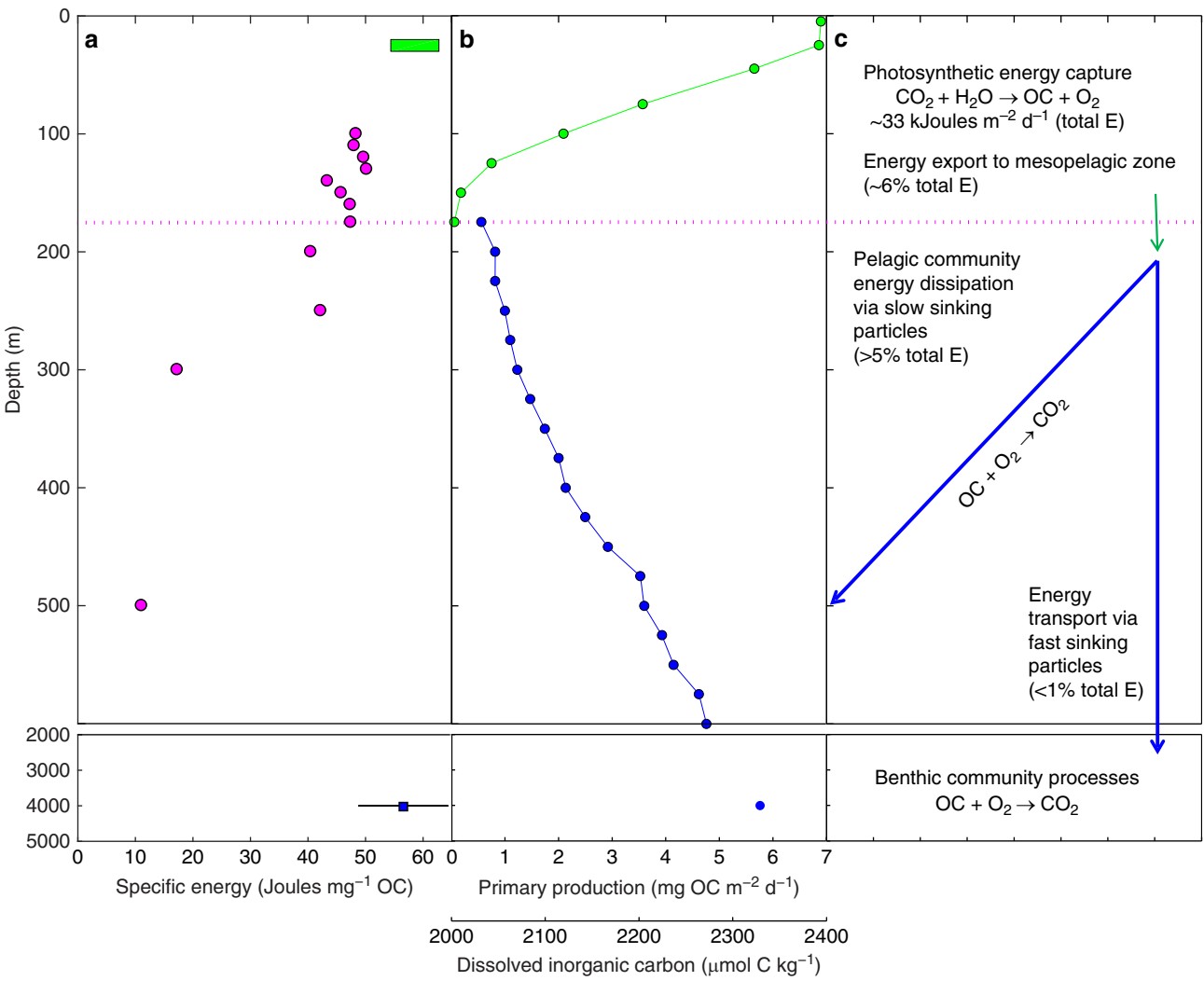

**Fig. 3** Energy capture, flux, and dissipation. **a** Carbon and energy flux processes showing specific energy of photosynthetic production (green bar), depth-dependent changes in energy of SPM as particles sink and age (magenta symbols) and mean ± s.d. specific energy for SPM collected at 4000 m (blue symbol). The horizontal magenta dotted line at 175 m depicts the photosynthetic compensation irradiance (see text for details). **b** Profiles of photosynthetic production (green symbols) and concentration of dissolved inorganic carbon (blue symbols) at Station ALOHA. **c** Schematic representation of the major processes controlling carbon and energy fluxes at Station ALOHA

reference depth (Fig. 1 and Table 1), indicates preferential BC preservation relative to OC.

Only one previous study reported BC in sinking particles[15]; our report is the first to quantify the BC flux profile in the upper water column. BC refers to a broad spectrum of refractory materials (e.g., graphite, soot, char; ref. [14]) that are formed during incomplete combustion of fossil fuels, wood, and other organic matter. BC is ubiquitous, and has been reported as a component of both dissolved and particulate matter in the ocean[18–21]. Because we used a conservative method to estimate BC in our samples (see Methods), our reported concentrations are probably lower bounds on BC concentration. The source(s) of marine BC are poorly known, but it appears that there are multiple components, some with radiocarbon ages of >20,000 years[22]. Regardless of BC sources and dynamics, the low mass- and OC-specific energies of SPM collected at 500 m are indicative of SPM that may be at or near the lower limit for further microbial decomposition.

**Nominal oxidation state of carbon and Gibbs energy of SPM.** LaRowe and Van Cappellen[23] have shown that the standard molar Gibbs energies of the oxidation half reactions for a variety

of organic compounds is inversely proportional to the nominal oxidation state of carbon (NOSC) and approaches a value of zero at NOSC >2.2. Although there are no direct measurements of oxygen in SPM, previous studies have estimated oxygen by difference between the total mass flux and all other major constituents (organic matter, carbonate, silicate[24]). If we assume a late summer opal ($SiO_2$) flux of 3 mg Si m$^{-2}$ d$^{-1}$ (ref. [25]), and assign oxidation numbers to H(+1), O(−2), N(−3), and P(+5), we derive a NOSC estimate of 2.19 for SPM collected at 500 m, which supports our hypothesis of an oxidized, energy-depleted organic matter pool. In addition, when comparing the enthalpies of combustion and Gibbs energies of combustion for a broad range of organic compounds under standard conditions, we found that nearly all of the values (83 of the 91 organic compounds) for the enthalpies and Gibbs energies agreed to within ±10% (Supplementary Fig. 3), a result that is consistent with the conclusion that aerobic metabolisms are enthalpy-driven with little contribution from entropy[10]. We further developed a multiple linear regression model to estimate the standard Gibbs energy of combustion of these same 91 organic compounds based on their elemental composition ratios (H:C, N:C, O:C, and S:C;

**Table 3 Mass, carbon, and energy fluxes of sinking particulate matter at 4000 m**

| Collection period | Winter[a] (4–11 Nov 2016) | Summer (4–11 July 2016) |
|---|---|---|
| Mass flux (mg m$^{-2}$ d$^{-1}$) | 15.62 ± 1.72 | 31.20 ± 2.93 |
| Carbon flux (mg m$^{-2}$ d$^{-1}$) | | |
| OC | 0.85 ± 0.02 | 2.41 ± 0.10 |
| IC | 0.88 ± 0.01 | 1.88 ± 0.09 |
| BC | 0.02 ± 0.00 | 0.06 ± 0.01 |
| Hydrogen flux (mg m$^{-2}$ d$^{-1}$) | 0.15 ± 0.01 | 0.40 ± 0.01 |
| OC:H:N:P (molar) | 150:318:15:1 | 205:409:20:1 |
| Energy flux (J m$^{-2}$ d$^{-1}$) | 46.81 ± 2.92 | 107.07 ± 6.05 |
| Specific energy | | |
| (J mg$^{-1}$ mass) | 3.00 ± 0.38 | 3.43 ± 0.38 |
| (J mg$^{-1}$ OC) | 55.17 ± 3.73 | 44.51 ± 3.11 |

[a]All values are reported as mean ±1 standard deviation

Supplementary Fig. 4). The result of this analysis can be used to provide an independent estimate of the Gibbs energy of combustion based on the elemental composition of organic matter. The standard Gibbs energy of combustion for SPM collected at 500 m estimated from our stoichiometric model is −21 J mg$^{-1}$ OC compared to an average of −31 J mg$^{-1}$ OC in the upper 100–175 m portion of the water column. The actual in situ Gibbs energy would depend on the activities of substrates and products, as well as on temperature and pressure. Consequently, the apparent recalcitrant nature of the mesopelagic SPM may be due to kinetic as well as thermodynamic constraints. While these approaches are only approximations, the results emphasize the importance of a comprehensive stoichiometric and energetic analysis of SPM.

Our data for the downward energy flux and specific energy of SPM are the first reports of their kind for the BCP. Previously, Platt and Irwin[26] reported proximate analyses, bulk C:N ratios, and energy estimates for the spring bloom of phytoplankton in St. Margaret's Bay, Nova Scotia. They concluded that the dry weight percent C and N of suspended particles could be used to estimate its energy content. We applied their model to our SPM samples where direct bomb calorimetric analyses have also been made. As expected, the Platt and Irwin[26] approximation yields an over-estimation of the measured energy content of SPM at 500 m, likely due to the fact that the chemical compositions and energy contents of mesopelagic SPM are different from those of living phytoplankton, and they change significantly over the 100–500 m region of the water column as particles sink and age, as documented herein.

**Abyssal energy fluxes**. Pioneering sediment trap studies in the deep sea have documented rapid, short-lived pulses of organic matter following the spring bloom of surface phytoplankton[27]. Distinct seasonal variations in the amount and composition of SPM collected at 4000 m have also been reported at Station ALOHA[9]. Based on a 13-year climatology, Karl et al.[9] documented a relatively low and nearly constant flux of OC, N, P, and biogenic silica (opal) that was overprinted by a prominent peak in export, especially for OC and N, for a brief period that they referred to as the summer export pulse. Changes in the bulk elemental, biochemical, and molecular composition of SPM during late summer were attributed to the rapid export of symbiotic nitrogen-fixing cyanobacteria in association with diatom hosts, and the summer export pulse was hypothesized to be triggered by a response of the surface plankton to daylength[9]. We collected additional deep-sea sediment trap samples in 2016 to measure the elemental stoichiometry and specific energy of SPM during winter and late summer periods (Table 3). As expected, the fluxes of total mass, OC, H, N, and P in winter were lower than in summer. The wintertime energy flux was also significantly lower (Table 3), with values that are ~2% of those measured at 100 m (Fig. 3). In contrast, the summertime energy flux at 4000 m is more than twice the wintertime value (Table 3). Whereas the mass-specific energy in both winter and summer is relatively low (3.00 and 3.43 J mg$^{-1}$) and similar in magnitude to the 300–500 m SPM, the OC-specific energy values for SPM collected at 4000 m during both the winter and late summer periods are higher than any measured in the water column and are on par with the estimates measured in phytoplankton blooms (Fig. 3 and Table 3). We resampled our abyssal SPM archive from Station ALOHA to measure the OC-specific energies for SPM collected in both winter and summer periods in 1998 and 2000. These results conform to the 2016 data, with wintertime values of 60.26 (s.d. = 6.77) and 64.13 (s.d. = 5.88) J mg$^{-1}$ OC and summertime values of 64.09 (s.d. = 10.48) and 51.21 (s.d. = 1.88) J mg$^{-1}$ OC for 1998 and 2000, respectively. The mean OC-specific energy content for all 4000 m SPM samples analyzed to date is 56.56 (s.d. = 7.79) J mg$^{-1}$ OC. This value is not significantly different from the mean value of 60.67 J mg$^{-1}$ OC reported above for SPM that is remineralized in the 100–500 m region of the upper water column and to the OC-specific energy stored during photosynthesis (ref. [17] Fig. 3a). Consequently, while abyssal environments appear to be limited with respect to the total flux of potential energy, the SPM that arrives there has a very high OC-specific energy, regardless of season.

## Discussion

All life processes on Earth, including human economic and social systems, exist within a complex network of energy flow[2]. In oceanic systems, photosynthesis is the primary source of both organic carbon and energy, but the subsequent pathways and mechanisms for carbon cycling and energy dissipation are less well understood. For example, in the present study, we observed that the elemental compositions and specific energy values for SPM collected in the upper mesopelagic zone (300–500 m) were fundamentally distinct from SPM collected at an abyssal (4000 m) depth. The relatively high OC-specific energy of SPM collected at 4000 m can be explained by several possible mechanisms that are not mutually exclusive. First, vertically segregated zones of new particle production deep within the water column could serve to repackage surface-derived organic matter, resulting in the flux of a smaller mass of more energy-replete particles[28]. This repackaging process may be facilitated by diel, vertically migrating zooplankton that transport fresh organic matter to depths of 800–1000 m at Station ALOHA[29]. In addition, the flux of relatively energy-replete SPM to 4000 m could be a result of the selective preservation of lipids[30] or other energy-rich organic compounds during transit to the deep sea. However, the relatively

energy-depleted SPM collected at 300 and 500 m (17.3 and 11.1 J mg$^{-1}$ OC, respectively) argue against lipid preservation with increasing water depth, at least for the upper portion of the mesopelagic zone. Finally, the mechanism which may best explain the OC and energy flux patterns that we observe for SPM, is the control by large, rapidly sinking particles that escape the combined processes of disaggregation, dissolution, solubilization, and microbial decomposition en route to the deep sea[31] (Fig. 3c). The relatively high total mass-to-OC ratios of the 4000 m SPM (~15; Table 3) are at least three times greater than those measured in near-surface waters, suggesting an important role for biogenic mineral ballast[32]. Indeed, ~50% of the SPM collected at the 300 m reference depth at Station ALOHA during the VERTIGO expedition in June 2004 had sinking speeds greater than 100 m d$^{-1}$ and >15% of the material was sinking at rates >820 m d$^{-1}$ (ref. [33]). This rapidly sinking and presumably energy-replete SPM would reach the seabed in less than 1 week. Previous studies at Station ALOHA[9] have documented the significant role of diatoms in controlling particle flux to the deep sea. A peak in opal and intact diatoms coincided with the summer export pulse of organic matter[9]. Herein, we also present evidence for a significant contribution of calcium carbonate with deep-sea IC representing nearly half of the total C flux (Table 3). These high proportions of opal and calcium carbonate, the two most important ballast biominerals[34], are not observed in the upper water column SPM, thereby providing strong evidence for a distinctive class of fast-sinking energy-replete particles in the deep-sea samples (Fig. 3 and Table 3). The 4000 m SPM is also characterized by low BC: OC ratios that are more similar to the euphotic zone SPM (100–175 m) than to the BC:OC ratios of the SPM collected deeper in the water column (300–500 m; Table 1). By our analysis, ~10% of the OC exported from the euphotic zone and ~0.4% of the total energy initially captured via photosynthesis reaches the abyssal benthic community at Station ALOHA while the remaining ~90% of the exported OC is remineralized to carbon dioxide en route to the seafloor (Fig. 3b). Predicted future states of a warmer, more acidic, and more nutrient-depleted NPSG[35,36] may alter the phytoplankton community structure and select against opal- and calcium carbonate-containing organisms[37,38]. Under this environmental scenario, the flux of carbon and energy to the deep sea will be reduced due to a decrease in the proportion of mineral-ballasted SPM. A disruption in food supply to the deep sea would have a significant effect on the structure and metabolism of abyssal benthic communities[39]. While we have focused our discussion on the role of OC and energy fluxes, other components of the SPM pool, such as ammonium and sulfide, might also contribute to the total energy flux[40,41] and should be considered. While the data presented in this study derive from only a single station in the NPSG, the flux attenuation patterns, role of ballast minerals and changes in organic matter stoichiometry of SPM with depth appear to be common features throughout large portions of the global ocean. Quantitative studies of these and other linkages between the C cycle and the pathways of biological and detrital energy flow will be required to develop a comprehensive understanding of the ocean's BCP and its role in C sequestration.

## Methods

**Energy capture via photosynthesis**. A previous study of light absorption by phytoplankton at Station ALOHA during 2006–2012 reported that the mean 0–200 m depth-integrated value in summer was 0.79 (s.d. = 0.19) mol quanta m$^{-2}$ d$^{-1}$, and approximately 20% lower (0.64 (s.d. = 0.16) mol quanta m$^{-}$ d$^{-1}$) in winter[6]. Assuming blue light ($\lambda = 475$ nm), we calculated absorbed energy ($E$) as $E = h\nu$, where $h$ = Planck's constant ($6.63 \times 10^{-34}$ J s$^{-1}$), $\nu$ = frequency defined as speed of light ($C$) divided by $\lambda$. This resulted in a photosynthetic energy capture of 33,054 J m$^{-2}$ d$^{-1}$ for the period of our upper ocean sediment trap experiment.

**SPM collection and processing**. Sinking particles were collected during the HOE-PhoR II expedition in September 2013 at Station ALOHA (22°45′N, 158°W)[42]. A free-drifting sediment trap array, identical to that described by Knauer et al.[43] consisting of eight individual particle interceptor traps (PITs) on PVC crosses positioned at twelve depths ranging from 100 to 500 m was employed. Prior to deployment, each PIT was filled with a filtered, buffered formalin brine solution consisting of filtered surface seawater containing 50 g l$^{-1}$ sodium chloride and 1% (vol/vol) formalin. The array was deployed over a 9.1-day period. Following recovery, individual PITs were capped and placed in a cool, dark area. The interface between the high density trap solution and the overlying seawater was marked, and the overlying seawater was removed to within 5 cm above the interface. All samples were passed through a 335 µm Nitex® screen to remove zooplankton and micronekton, as previously described[12]. Each sample was vacuum filtered onto a tared 47 mm diameter, 0.8 µm porosity polycarbonate membrane. In order to remove salts, the filtered samples were rinsed six times, each with 5 mL of deionized water (DI), allowing each rinse to re-suspend the particles before filtration. The filters were stored frozen until further analysis. The filters were subsequently dried at 60 °C for 12 h, stored in a dessicator and periodically weighed to achieve constant weight in order to calculate mass flux from PIT collection area (0.0039 m$^2$) and deployment time (9.1 days). Each filter was then placed into a vial containing 5 mL of DI, and the particles were continuously mixed on a shaker table for 1 h. Finally, any particulate matter remaining on the filter was rinsed into the vial with additional DI, and the entire volume was evaporated in a SpeedVac®, leaving the dried particles behind. The sediment was pulverized and dried at 60 °C for 8 h. This powdered material was used for all subsequent analyses.

**Elemental analysis**. Various amounts of sample from each depth were sealed in tared tin cups and weighed on a microbalance. The C, H, and N contents were determined using an Exeter Analytical Elemental Analyzer using a combustion temperature of 1020 °C. Acetanilide was used as a standard and the Hawaii Ocean Time-series (HOT) plankton sample was used as a quality control reference material between sample runs. This value of C represents total C (TC) consisting of inorganic (IC), black (BC), and organic (OC) components. Aliquots of each sample were also used to measure IC and BC. For IC (calcium carbonate) determinations, subsamples (~0.4 mg) were placed into a glass vial and sealed with a gas-tight stopper. Phosphoric acid (200 µl of an 8.5% vol/vol solution) was added and the sample was incubated for 1 h after which time the entire headspace of the vial was flushed through an infrared detector to measure total carbon dioxide by peak integration. Various volumes of a 410 ppm carbon dioxide in air standard were used for calibration and reagent grade calcium carbonate was also analyzed as a check standard. For BC determinations, we employed the CTO-375 method of Gustafsson et al.[44]. A recent laboratory intercomparison of BC quantification in 12 different source materials using 7 different analytical methods[45] provides a comprehensive evaluation of the strengths, weaknesses, and limitations of each method. The method that we selected specifically targets the most highly condensed aromatic structures (e.g., soot), and therefore provides a lower bound on total BC in our samples. Our analysis ($n = 17$ samples) of National Institute of Standards and Technology marine sediment (SRM 1941b), which represents BC in a sediment matrix similar to our SPM, yielded the following: BC = 0.58% (s.d. = 0.02) of total dry weight, 3.09% OC of total dry weight and 18.70% BC as a % of total OC. OC in our study is reported as: OC = TC − [IC + BC]. Particulate P content was determined by high-temperature ashing and colorimetric analysis[12]. Elemental fluxes were estimated from the respective analyses, PIT collection area, and deployment time. Replicate PITs deployed on the same array were also processed for C and N using procedures developed in the HOT program that employed direct filtration of particles onto glass fiber filters[12] which are incompatible for measurements of mass and total energy content. The two independent methods for C and N were not significantly different.

**Total energy content**. Energy content was determined by measuring the amount of heat released by combustion in an oxygen bomb calorimeter (Parr Instrument Company Semimicro Calorimeter) following the manufacturer's recommended procedures with only slight modifications. This instrument is a constant-volume calorimeter designed specifically for small sample sizes ranging from 25 to 200 mg. A press was utilized to prepare a compact pellet of each sample prior to combustion. Duplicate or triplicate (dependent on amount of available sample) pellets (5–9 mg each) from each depth were measured in this study. Due to our sample size limitation and the instrument's requirement of at least a 25 mg sample, an exact amount of benzoic acid (heat of combustion 26,435 J g$^{-1}$) was added to each sample. The addition of benzoic acid ensured complete combustion of the sample. The ideal weight for spike addition was determined to be ~30 mg by testing numerous amino acids and plankton samples in the 5–9 mg range. Replicate samples containing variable benzoic acid-to-sample ratios (from 2 to 7) returned identical estimates of the sample caloric content within the approximately 3–6% reproducibility of the bomb calorimetric assay. Pure benzoic acid pellets were used to determine the heat capacity of the calorimeter-standardization prior to sample analysis, and were also used as check standards during each sample run. Formation of nitric acid[46,47] was routinely assessed by postcombustion rinsing of the bomb chamber with DI and titration with 0.0725N sodium carbonate. Assuming that all

acid produced was nitric acid, we determined that, on average, the acid correction was <4% of the total heat of combustion for our samples, so it was ignored in our subsequent calculations. Paine[48] previously reported an endothermy value of $-0.573$ J mg$^{-1}$ calcium carbonate. While our SPM samples contained various amounts of calcium carbonate (i.e., deep-sea samples had IC:OC molar ratios ~1:1), the use of benzoic acid as a filler decreased the IC:OC ratios to below the threshold where interference (endothermy) was significant in the determination of total energy content of our samples. Consequently, no correction was required for calcium carbonate endothermy.

## Data availability

Data files are available at hahana.soest.hawaii.edu/GBMF/index.html under project "Coupling carbon and energy fluxes in the North Pacific Subtropical Gyre".

## Code availability

The Matlab® code for the thermodynamic calculations is available at hahana.soest.hawaii.edu/GBMF/index.html under project "Coupling carbon and energy fluxes in the North Pacific Subtropical Gyre".

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

## Acknowledgements

We thank Blake Watkins, Tara Clemente, Karin Björkman, Dan Sadler, Lance Fujieki, and Lisa Lum for their assistance in the field and in our shore-based laboratories. This research was supported by the National Science Foundation (EF-0424599; DMK), the Gordon and Betty Moore Foundation (#3794; DMK) and the Simons Collaboration on Ocean Processes and Ecology (SCOPE #329108; DMK).

## Author contributions

D.M.K. and E.G. designed and conducted the field study. E.G. processed and analyzed the sediment trap samples. E.A.L. wrote and executed the Matlab code. D.M.K. wrote the initial draft of the paper. D.M.K., E.G., R.M.L. and E.A.L. discussed and interpreted the data, and contributed to preparation of the final paper.

## Additional information

**Competing interests:** The authors declare no competing interests.

