## [Peer Review File · Nature Communications]

Reviewers' comments:

Reviewer #1 (Remarks to the Author):

I found this paper to be very novel, of high importance and well presented. It is novel because of the concept of tracking energy density, not simply C, mass or other fluxes on sinking particles out of the surface ocean. They reach two main conclusions: 1) that in the upper 100-500m, the more recently produced sinking mater is selectively and rapidly remineralized and 2) despite this, what arrives at the seafloor looks much more like the surface source, i.e. fresh and high energy content. This finding is shown here better than in other papers by focusing on novel energy density results presented for the first time for traps that I know of.

So I urge publication, but feel some small changes are warranted, as this will be an important paper for future reference. First, some caveats are needed. This is one set of observations in Sept. 2013 100-500m and we know the character of sinking particles varies seasonally and spatially around the world's oceans. So it is hard to know how well this data applies elsewhere- this needs to be brought out. Likewise, they compare shallow flux results to deep traps from other years, and they find at depth fresher/higher energy material year round and higher fluxes after the summer export peak (please define first use SEP). There is a long history of finding fresh material on the seafloor that they should reference, ranging from early fallout radionuclides found by C. Osterberg et al. 1963? Nature, and recognized by many biologists, in 1980's see Theil, Lochte and others talking about fresh detritus on the seafloor. The point is that this energy density being high in the deep trap is consistent with prior studies. The new finding is that the majority of the flux, and energy, is lost at shallow depths. So it is a two (or more) component system as shown, with a small fraction of fast sinking high energy density particles, but they are not the first to talk about this fast sinking fresh material on the seafloor. Obviously this paper leaves a big opening for future work with similar energy density measurements, to see if this is universal? unique to the ALOHA SEP? (not likely), and someone will need to make a better connection in space/time between shallow samples and deep traps. So some brief suggestions about future work would be appropriate.

Couple minor points-

102- define euphotic zone, as there are Chl and other pigments deeper than 100m at ALOHA, the depth they call the euphotic zone (maybe referring to depth above which XX% of NPP takes place?).

137- The BC story is interesting, but while percentage increases from 100 to 500m, this is still a minor fraction of the total C.

189- In terms of 3 possible mechanisms, first is diel migration, but then wouldn't there be a flux peak at the depth of migration (if zooplankton are bringing down lots of fresh C?), so to me this reason is not supported by prior flux profile data. I agree that lipids don't work so final mechanism is good, as is reference to Trull showing 15% of material sinking very fast (fits with what I said before, that no one doubts that some small fraction is sinking fast and must therefore be fresh). Methods- seem OK, but I hope at least one reviewer is more familiar with methods for total caloric content than this reviewer, as it's a key measurement and not done on trap samples like this before, so the whole paper hinges on this new method, but I have to trust the authors and co-authors on this.

Fig. 2. The jump in energy density for POC seems to be a step function between 250 and 300m rather than a gradual shift. Any comments on why that might be?

Thanks for including data tables. I'm sure others will be interested. Also such high resolution trap data in the vertical is unique and good to be sharing for all of the elements and energy densities.

In short- good work! Minor improvements would help put this in better context of prior ideas, but overall it is important to get this interesting and novel work published.

Reviewer #2 (Remarks to the Author):

This manuscript reports how sinking particulate matter (SPM) varies with depth at the ALOHA station in the North Pacific Subtropical Gyre. The authors conclude that microbial remineralization of organic carbon (OC) in SPM changes its composition, leaving an energy-depleted mass of OC in SPM in the top 500 m of the water column. By contrast, SPM reaching 4000 m water depth are energy-replete, which is attributed to rapid sinking. The authors conclude by noting that as the climate changes, the upper-ocean biological response will be to produce less of this fast-sinking, energy replete OC to the seafloor, damaging benthic communities. In a one sentence review, this manuscript would be well-served by a more rigorous accounting and discussion of energy and the energy content of organic matter.

The results of this study are discussed in terms of energy, but the authors are rather sloppy in their usage of this term. This begins with nomenclature and units. Throughout the manuscript the authors refer to energy densities, but the reported units are calories per mass. This is specific energy (for instance, specific heat capacities are in units of $J / (K kg)$). Energy density is energy per volume, J / m^3 , a unit commonly used to describe different types of fuel. Perhaps the authors could use the term 'energy intensity' as it in physics: energy per area per time, which would might dovetail nicely with their calculated energy of photons (which are commonly reported as intensities). Also, the SI unit for energy is Joule.

It's not entirely convincing that 'heat' can be used to quantify the energy content of organic matter that is being consumed by microorganisms. This is because "heat," as the authors are using it, refers to enthalpy of combusting organic compounds. This energy that organisms obtain by oxidizing organic matter is quantified using Gibbs energies. It is true that enthalpies of combustion can be converted into enthalpies of formation and that enthalpies of formation can be used to calculate part of the Gibbs energies of reactions, but these values are numerically – and conceptually – distinct. For example, the enthalpy of combustion of glucose(cr) = -2805 kJ/mol; the standard enthalpy of formation of glucose(cr) = -1273 kJ/mol; and the standard Gibbs energy of glucose(cr) = -908 kJ/mol. Taking environmental conditions such as temperature, pressure and composition into account would further augment this Gibbs energy value. An organism does not catalyze a reaction to gain energy because of its enthalpy of combustion, but because of its Gibbs energy content.

The data that the authors generated actually can be used to estimate the amount of Gibbs energy that microorganisms could get by oxidizing organic matter (see LaRowe and Van Cappellen, 2011). Briefly, the half reaction describing the complete oxidation of organic compounds is inversely related to the oxidation state of that carbon: overall, the microbial oxidation of oxidized carbon compounds yields more energy than reduced ones per mole of carbon. Although it is an incomplete calculation since the authors did not determine the stoichiometric amount of oxygen in their organic samples, the calculated oxidation state of OC in their samples decreases with depth (see plot), meaning that microorganisms are selectively oxidizing the more energy-rich oxidized organic compounds and functional groups as the SPM sinks. This more or less confirms the authors results based on enthalpies of combustion. However, the organic carbon in the 4000m samples are even more reduced, suggesting that either the oxidized organic compounds have been even more selectively oxidized than the samples in the top 500 m (and are therefore not as energy-replete as the authors suggest) or that different types of organic compounds are associated with the ballast that makes them sink so fast. Again, this is an incomplete analysis since the molar ratio of oxygen to carbon is not given and would influence values of the oxidation state of OC and therefore the Gibbs energies of its oxidation. However, there would have to be a very large stoichiometric transition in O content to dramatically alter the oxidation state of this organic carbon.

Furthermore, it would be helpful if the authors could better connect the energetics of a photon to the enthalpies of combustion of their organic samples. Photosynthetic organisms use electromagnetic energy to liberate electrons from a donor, commonly water. Both the water-splitting reaction and the overall reaction of carbon fixation, $6\text{CO}_2 + 6\text{H}_2\text{O} \rightarrow \text{C}_6\text{H}_{12}\text{O}_6 + 6\text{O}_2$, actually have a positive enthalpies: they are endothermic and therefore absorb heat ("light" does not have an enthalpy of combustion or formation so it does not enter into such a reaction). This is why when the OC is combusted, effectively the reverse of the carbon fixation reaction, it releases heat. Currently, the authors label their reactions with the word 'heat' on the right side as if they all release heat, but this isn't so. Furthermore, "heat" is not a reactant or product in a chemical reaction, but a thermodynamic quantity related to how the internal energy of a system varies at constant pressure. It should never appear in a chemical reaction. Finally, the heat associated with a reaction does not fully determine its direction – enthalpy is only a statement of the 1st law of thermodynamics, the conservation of energy. It is the Gibbs energy that predicts directionality by incorporating the second law as well. Just because many thermodynamic quantities have the same units, does not make them so easily comparable – enthalpies of combustion, formation and reaction and Gibbs energies, Helmholtz energies and internal energy all have units of Joules per mol, but they are very different from one another.

The following are specific points about the manuscript that require attention as well:

It's noted that ATP and NADPH are used to reduce CO_2 , producing heat in the process. As noted above, the conversion of CO_2 into glucose absorbs heat and since NADPH and ATP participate in cycles, any heat released from their oxidation and hydrolysis, respectively, is canceled out by their subsequent regenerative synthesis. If the authors are referring to energy that is used inefficiently, then that should be stated. Besides, it's not always just heat – all life exports entropy, another form of energy.

Line 55: Several regions have been referred to as the world's largest biome: global marine sediments, oceanic basement, and the terrestrial subsurface, all of which are larger than an ocean gyre. Just say that it's one of the largest.

Line 85: black carbon is not highly oxidized. The average nominal oxidation state of carbon in polyaromatic hydrocarbons is < 0 , more reduced than glucose, which is 0. Even if black carbon were pure graphite, it's oxidation state would be 0.

Line 92 this is a confusing way to report the stoichiometry of organic matter since O typically stands for oxygen in this context. Drop the O and say that this is the stoichiometry of CHNP in organic carbon; Furthermore, this is a rather incomplete analysis of the stoichiometry of natural organic matter, which includes a fair amount of oxygen.

Lines 142-144: recalcitrance of organic carbon is a function of the ecosystem; fullerenes, black carbon and kerogens can be consumed by microorganisms given the right conditions. After all, the radio carbon age of black carbon in seawater that is noted is 20,000 years old, not radio-carbon dead.

Line 163: define SEP; don't make the reader sift through another paper to define an acronym

Line 184: wrong reference

It's quite a stretch to use data from one spot in the ocean to extrapolate how fluxes of particulate organic carbon will change in response to climate change. This requires more data to substantiate.

Table 1: the last four columns should say that these are the fluxes of these elements in the organic matter in SPM, if that's what they are.

Figure 1: put some space between the figures.

The caption is vague; display items should be able to stand alone with their captions (i.e., define the acronyms); These are flux profiles and mass... of sinking particulate matter as a function of depth at the ALOHA station...;

If you're going to dump model parameters in the caption, you have to show the related equation somewhere and define what the parameters are.

Figure 2 : This is not the "total energy content" It is, I think, the enthalpies of combustion of organic carbon in SPM (total energy would include gravitational, kinetic, nuclear, etc.); same comment as for Fig 1 regarding model parameters and acronyms;

Fig. 2b needs more of an explanation; flux's don't have 'arbitrary units'; perhaps these are non-dimensionalized fluxes?

Figure 3c: this caption needs to be more specific; "this panel is a schematic of..."

As noted above, these are not chemical reactions and not all of them release heat.

Response to Reviewer Comments

We thank the Editor and two external reviewers for the time and effort they have invested. We are especially appreciative of the comments and constructive criticisms of our work which, we believe, have greatly improved our presentation. Below (in bold font) we provide our responses and detail the major changes that have been made in our revised manuscript.

Reviewer #1 (Remarks to the Author):

I found this paper to be very novel, of high importance and well presented. It is novel because of the concept of tracking energy density, not simply C, mass or other fluxes on sinking particles out of the surface ocean. They reach two main conclusions: 1) that in the upper 100-500m, the more recently produced sinking mater is selectively and rapidly remineralized and 2) despite this, what arrives at the seafloor looks much more like the surface source, i.e. fresh and high energy content. This finding is shown here better than in other papers by focusing on novel energy density results presented for the first time for traps that I know of.

We thank the reviewer for this very favorable assessment of our work.

So I urge publication, but feel some small changes are warranted, as this will be an important paper for future reference. First, some caveats are needed. This is one set of observations in Sept. 2013 100-500m and we know the character of sinking particles varies seasonally and spatially around the world's oceans. So it is hard to know how well this data applies elsewhere- this needs to be brought out.

This is an excellent point. In the revised manuscript we are careful not to draw any conclusions beyond our own well-studied open ocean site, Station ALOHA. Nevertheless, the general attrition of particle flux versus depth (often modeled by the “so-called” Martin curve) and the importance of ballast minerals appear to be universal phenomena that have been reported from many diverse marine ecosystems. Future studies of the coupling of mass and energy fluxes will obviously need to be conducted. We hope that our research will stimulate those efforts.

Likewise, they compare shallow flux results to deep traps from other years, and they find at depth fresher/higher energy material year round and higher fluxes after the summer export peak (please define first use SEP). There is a long history of finding fresh material on the seafloor that they should reference, ranging from early fallout radionuclides found by C. Osterberg et al. 1963? Nature, and recognized by many biologists, in 1980's see Theil, Lochte and others talking about fresh detritus on the seafloor. The point is that this energy density being high in the deep trap is consistent with prior studies. The new finding is that the majority of the flux, and energy, is lost at shallow depths. So it is a two (or more) component system as shown, with a small fraction of fast sinking high energy density particles, but they are not the first to talk about this fast sinking fresh material on the seafloor. Obviously this paper leaves a big opening for future

work with similar energy density measurements, to see if this is universal? unique to the ALOHA SEP? (not likely), and someone will need to make a better connection in space/time between shallow samples and deep traps. So some brief suggestions about future work would be appropriate.

Thank you for reminding us about the early particle flux papers that very clearly documented “phytodetritus” (fresh organic matter) at the deep seabed. We have now added a reference to those pioneering studies (Billett et al. *Nature*). We have also added a few brief comments on future work, specifically on the urgent need for measurements of oxygen content of SPM and on quantitative estimation of SPM sinking rates in the open sea.

Couple minor points-

102- define euphotic zone, as there are Chl and other pigments deeper than 100m at ALOHA, the depth they call the euphotic zone (maybe referring to depth above which XX% of NPP takes place?).

The reviewer is correct. The photosynthetic compensation irradiance (the irradiance at which net photoautotrophic carbon assimilation is zero over a 24-hr period) at Station ALOHA is 0.054 mol photons m⁻² d⁻¹, or about 0.11% of surface 400-700 nm radiation (Laws et al. 2014, *Deep-Sea Research* 93: 35-40). This corresponds to a euphotic zone depth of ~175 m. Less than 10% of the total euphotic zone primary production occurs between 100-175 m (Letelier et al. 1996, *Deep-Sea Research II* 43: 467-490).

137- The BC story is interesting, but while percentage increases from 100 to 500m, this is still a minor fraction of the total C.

We agree with the reviewer that the presence of black carbon (BC) in sinking particulate matter is an interesting observation. As we describe in the methods section of our paper, we used a method for BC analysis that is “conservative” when compared to other methods that have been reported in the literature. The seven most commonly used methods for BC analysis were compared using a variety of sample materials, including soils and a marine sediment (SRM1941b from the U.S. National Institute of Standards and Technology). The CTO-375 method, which we employed in our study, returned the lowest BC concentrations by factors of 4-10 of all methods tested, probably because it targets only highly condensed BC (Hammes et al. 2007, *Global Biogeochemical Cycles* 21). Consequently, our reported BC concentrations probably represent lower bounds on BC in our sediment trap collected materials. In the future, we plan to use several of the different methods outlined in this intercomparison to better constrain and quantify the BC content. We believe that the presence of BC in oceanic SPM is potentially a very important part of BCP of the open sea.

189- In terms of 3 possible mechanisms, first is diel migration, but then wouldn't there be a flux peak at the depth of migration (if zooplankton are bringing down lots of fresh C?), so to me this reason is not supported by prior flux profile data. I agree that lipids don't work so final mechanism is good, as is reference to Trull showing 15% of material sinking very fast (fits with

what I said before, that no one doubts that some small fraction is sinking fast and must therefore be fresh).

We agree with the reviewer that the two component model with a small portion of fast sinking, energy-replete material arriving in the deep-sea is the most tenable hypothesis. In future field studies we hope to collect fast sinking (only) particles from a variety of depths using either a Marine Snow Catcher (MSC; Riley et al. 2012, *Global Biogeochemical Cycles* 26) or a 4-stage elutriation system (Peterson et al. 2005, *Limnology and Oceanography Methods* 3: 520-532), or both, in conjunction with our conventional sediment trap array. This will provide sample materials from both the slow-sinking and rapidly-sinking components of the particulate matter pools for subsequent analysis.

Methods- seem OK, but I hope at least one reviewer is more familiar with methods for total caloric content than this reviewer, as it's a key measurement and not done on trap samples like this before, so the whole paper hinges on this new method, but I have to trust the authors and co-authors on this.

We thank the reviewer for his/her trust in our analytical procedures for measurement of enthalpy by bomb calorimetry. We stand by our reported methods and data.

Fig. 2. The jump in energy density for POC seems to be a step function between 250 and 300m rather than a gradual shift. Any comments on why that might be?

The reviewer asks an excellent question. There are several possible explanations, but we believe this is a manifestation of very rapid remineralization of slow-sinking organic matter in the lower euphotic and upper mesopelagic zones leaving behind much more refractory, oxidized organic matter. We have recently conducted a field experiment with replicated, paired "live" (unpreserved) and formalin-preserved sediment traps deployed at fairly high resolution throughout the 100-500 m region of the water column to evaluate and characterize both the sinking particles that enter the traps at time zero, and the *in situ* remineralization potential over a week-long incubation period. The experiment was successfully completed and samples are currently in various stages of analysis for elemental and energy fluxes. Some additional comments regarding the nominal oxidation state of carbon (NOSC) as it relates to the thermodynamics of organic matter degradation are included in our response to Reviewer #2.

Thanks for including data tables. I'm sure others will be interested. Also such high resolution trap data in the vertical is unique and good to be sharing for all of the elements and energy densities.

Thanks for this comment. We hope that others will also have interest in this study and will use these observations as a foundation for future field work.

In short- good work! Minor improvements would help put this in better context of prior ideas, but overall it is important to get this interesting and novel work published.

Thank you for your helpful comments and general encouragement.

Reviewer #2 (Remarks to the Author):

This manuscript reports how sinking particulate matter (SPM) varies with depth at the ALOHA station in the North Pacific Subtropical Gyre. The authors conclude that microbial remineralization of organic carbon (OC) in SPM changes its composition, leaving an energy-depleted mass of OC in SPM in the top 500 m of the water column. By contrast, SPM reaching 4000 m water depth are energy-replete, which is attributed to rapid sinking. The authors conclude by noting that as the climate changes, the upper-ocean biological response will be to produce less of this fast-sinking, energy replete OC to the seafloor, damaging benthic communities. In a one sentence review, this manuscript would be well-served by a more rigorous accounting and discussion of energy and the energy content of organic matter.

We thank the reviewer for this helpful assessment of our manuscript. Below we discuss specific changes and substantive improvements to the “accounting and discussion of energy and the energy content of organic matter.” Upon receipt of these reviews, I discussed this matter with a long-time colleague, Edward A. Laws, who has a rather unique scientific background. Laws was trained and received his Ph.D. degree in chemical physics before moving into the field of marine sciences. He is currently Professor of Environmental Sciences at Louisiana State University and has published extensively on observations and models of the biological carbon pump. During the summer, Ed and I discussed the current manuscript and, in particular, the constructive comments that were provided. He made significant contributions and is now listed as a co-author of the revised manuscript.

The results of this study are discussed in terms of energy, but the authors are rather sloppy in their usage of this term. This begins with nomenclature and units. Throughout the manuscript the authors refer to energy densities, but the reported units are calories per mass. This is specific energy (for instance, specific heat capacities are in units of J / (K kg)). Energy density is energy per volume, J / m³, a unit commonly used to describe different types of fuel. Perhaps the authors could use the term ‘energy intensity’ as it in physics: energy per area per time, which would might dovetail nicely with their calculated energy of photons (which are commonly reported as intensities). Also, the SI unit for energy is Joule.

We thank the reviewer for this important correction and for the suggestion of the use energy intensity. Typically in the marine photochemistry/photobiology and climatology literature, light energy is reported either as irradiance (radiant flux density), reported as quanta m⁻² s⁻¹ or W m⁻², or as photosynthetic photon flux density (PPFD; reported in energy units (Joules m⁻² d⁻¹ or W m⁻²) or in light units (mol photons m⁻² d⁻¹). Example publications include Ge et al. 2011, “Dynamics of photosynthetic photon flux density (PPFD) and estimates in coastal northern California,” *Theoretical and Applied Climatology* 105: 107-118 and Foyo-Moreno et al. 2017, “A new conventional regression model to estimate hourly photosynthetic photon flux density under all sky conditions,” *International Journal of Climatology* 37 (suppl. 1) 1067-1075. Following additional literature research on the proper terminology for our study, we came across a recent paper on “Frequency

dependent power and energy flux density equations of the electromagnetic wave” by Muhibbullah et al. (2017, *Results in Physics* 7: 435-439). In this paper, the authors make the distinction between energy density (Joules m⁻³) and energy flux density (Joules m⁻²) which appear to be different terms/units than photon flux density. Rather than introduce any confusion into our discipline about our own data, we have decided to adopt the terms “energy flux” in reference to the measured energy of SPM collected in sediment traps (Joules m⁻² d⁻¹) and “specific energy” in reference to energy per mass (Joules mg⁻¹ mass or mg⁻¹ OC). All energy data are now expressed in the SI unit, Joule. The use of energy flux is now analogous to mass and element flux so this should avoid any confusion.

It’s not entirely convincing that ‘heat’ can be used to quantify the energy content of organic matter that is being consumed by microorganisms. This is because “heat,” as the authors are using it, refers to enthalpy of combusting organic compounds. This energy that organisms obtain by oxidizing organic matter is quantified using Gibbs energies. It is true that enthalpies of combustion can be converted into enthalpies of formation and that enthalpies of formation can be used to calculate part of the Gibbs energies of reactions, but these values are numerically – and conceptually – distinct. For example, the enthalpy of combustion of glucose(cr) = -2805 kJ/mol; the standard enthalpy of formation of glucose(cr) = -1273 kJ/mol; and the standard Gibbs energy of glucose(cr) = -908 kJ/mol. Taking environmental conditions such as temperature, pressure and composition into account would further augment this Gibbs energy value. An organism does not catalyze a reaction to gain energy because of its enthalpy of combustion, but because of its Gibbs energy content.

We thank the reviewer for correctly stating that we are reporting enthalpy values of the sinking organic matter as determined by bomb calorimetry. We view our reported energy fluxes as upper bounds or potential energy for the ecological system, and the corresponding specific energy estimates as predictors of the potential to support subsequent microbial metabolism and growth as particles sink and age. The attrition of energy flux with depth and the decrease in specific energy are direct manifestations of selective organic matter consumption during the decomposition process. Ecosystems like those under consideration in our study are thermodynamically open systems so the enthalpy and specific energy estimates may be more complex than the model we present. However, as noted by reviewer #1, these are novel data and concepts that have not been previously applied to the ocean’s biological carbon pump. We believe this is a good start on incorporating these concepts into studies of the biological carbon pump. We agree with the reviewer that we need to be careful and precise with regard to terminology.

The data that the authors generated actually can be used to estimate the amount of Gibbs energy that microorganisms could get by oxidizing organic matter (see LaRowe and Van Cappellen, 2011). Briefly, the half reaction describing the complete oxidation of organic compounds is inversely related to the oxidation state of that carbon: overall, the microbial oxidation of oxidized carbon compounds yields more energy than reduced ones per mole of carbon. Although it is an incomplete calculation since the authors did not determine the stoichiometric amount of oxygen in their organic samples, the calculated oxidation state of OC in their samples decreases with depth (see plot), meaning that microorganisms are selectively oxidizing the more energy-rich oxidized organic compounds and functional groups as the SPM sinks. This more or less confirms

the authors results based on enthalpies of combustion. However, the organic carbon in the 4000m samples are even more reduced, suggesting that either the oxidized organic compounds have been even more selectively oxidized than the samples in the top 500 m (and are therefore not as energy-replete as the authors suggest) or that different types of organic compounds are associated with the ballast that makes them sink so fast. Again, this is an incomplete analysis since the molar ratio of oxygen to carbon is not given and would influence values of the oxidation state of OC and therefore the Gibbs energies of its oxidation. However, there would have to be a very large stoichiometric transition in O content to dramatically alter the oxidation state of this organic carbon.

We thank the reviewer for this important discussion on the oxidation state of carbon as it relates to our measurements of enthalpy and the flux of energy to the deep sea, and for pointing out the very interesting LaRowe and Van Cappellen paper (2011, *Geochimica et Cosmochimica Acta* 75: 2030-2042). As the reviewer states, our stoichiometric analysis of sinking particulate matter is incomplete. While we do report OC, H, N, P and total mass in sinking particulate matter (a more complete analysis than is typically reported, as noted by reviewer #1), we lack quantitative data on oxygen. This is a general problem in marine/soil biogeochemistry. Traditional methods for measurement of the oxygen content of organic materials such as dry/wet oxidation, destructive chlorination and hydrogenation (Elving and Ligett 1944, *Chemical Reviews* 34: 129-156) or the method using neutron activation (Veal and Cook 1962, *Analytical Chemistry* 34: 178-184) have rarely, if ever, been used in aquatic sciences. For the past several decades, commercial elemental analyzers (PerkinElmer, Isomass, Leco, Exeter, and Horibe, to name a few manufacturers) have become the method of choice for determinations of C, H and N in organic materials. These systems use high-temperature oxidation with O₂ followed by detection of gaseous products, either in their oxidized form or, in the case of N, following subsequent reduction to N₂ (the so-called Dumas method). Ironically, many of these manufacturers also sell O/N analyzers (e.g., Horibe EMGA-920 or Leco RO-478) that could be used to quantify oxygen in organic matter by high-temperature pyrolysis in a He carrier gas followed by detection of CO and CO₂ by non-dispersive infrared analysis. I searched the oceanographic literature and found only a single paper that reports both organic C and O for particles in seawater. Chen et al. (1996, *Marine Chemistry* 54: 179-190) present a comprehensive analysis of the stoichiometry of suspended (not sinking) particulate matter of the western North Pacific. Their particulate organic oxygen (POO) was measured using a commercial Leco RO-478 analyzer. C:H:N:O molar stoichiometry from 68 stations in the East China Sea, Sea of Japan and Philippine Sea averaged: 7.69 (±0.33): 8.73 (±1.62): 1.0: 4.12 (±0.64). All other organic oxygen concentrations for marine suspended/sinking particulate matter have been estimated by difference after the measured components have been subtracted from the total mass (e.g., Honjo 1980, *Journal of Marine Research* 38: 53-97). As lamented by Elving and Ligett (1944), “this places the sum of all errors on to the oxygen estimate.”

Nevertheless, we have used our reported fluxes to calculate the mean oxidation state of sinking organic C as suggested by the reviewer, and compared our results to the thermodynamic analysis in the LaRowe and Van Cappellen paper. In our analysis, the flux of oxygen associated with sinking particles was equated to the difference between the total flux and the sum of the fluxes of the other components on the assumption that the

inorganic C (IC) flux was calcium carbonate (CaCO_3) and the silica flux was opal (SiO_2). These are reasonable assumptions. Then oxidation numbers were assigned to H(+1), O(-2), N(-3) and P(+5). Based on these assigned oxidation numbers and the associated fluxes, we estimated the oxidation number of OC. With the exception of one “outlier” at 110 m (carbon oxidation number = 2.28), the mean and SD OC oxidation number (1.48 ± 0.12) was relatively constant above the photosynthetic compensation depth (see Figure below). However, from 175-500, the OC oxidation number systematically increases from a value of 1.16 (175 m) to 2.77 (500 m). Based on the thermodynamic analysis presented by LaRowe and Van Cappellen (2011, their Figure 3a), the standard molar Gibbs energy of the oxidation half reaction of organic C would approach zero at a nominal oxidation state of 2.2. This analysis supports our conclusion that the organic matter collected at the 500 m sediment trap may be unable to support microbial growth based on thermodynamic considerations. We have added a brief summary of this analysis to the revised manuscript. We thank the reviewer for suggesting this analysis which will now be the focus of future field studies where we will endeavor to obtain direct measurements of particulate organic O.

Furthermore, it would be helpful if the authors could better connect the energetics of a photon to the enthalpies of combustion of their organic samples. Photosynthetic organisms use electromagnetic energy to liberate electrons from a donor, commonly water. Both the water-splitting reaction and the overall reaction of carbon fixation, $6\text{CO}_2 + 6\text{H}_2\text{O} \diamond \text{C}_6\text{H}_{12}\text{O}_6 + 6\text{O}_2$, actually have a positive enthalpies: they are endothermic and therefore absorb heat (“light” does not have an enthalpy of combustion or formation so it does not enter into such a reaction). This is why when the OC is combusted, effectively the reverse of the carbon fixation reaction, it releases

heat. Currently, the authors label their reactions with the word ‘heat’ on the right side as if they all release heat, but this isn’t so. Furthermore, “heat” is not a reactant or product in a chemical reaction, but a thermodynamic quantity related to how the internal energy of a system varies at constant pressure. It should never appear in a chemical reaction. Finally, the heat associated with a reaction does not fully determine its direction – enthalpy is only a statement of the 1st law of thermodynamics, the conservation of energy. It is the Gibbs energy that predicts directionality by incorporating the second law as well. Just because many thermodynamic quantities have the same units, does not make them so easily comparable – enthalpies of combustion, formation and reaction and Gibbs energies, Helmholtz energies and internal energy all have units of Joules per mol, but they are very different from one another.

We thank the reviewer for these important comments on Gibbs energy and enthalpy. We have removed “heat” and “light” from all reactions and discussion.

The following are specific points about the manuscript that require attention as well:

It’s noted that ATP and NADPH are used to reduce CO₂, producing heat in the process. As noted above, the conversion of CO₂ into glucose absorbs heat and since NADPH and ATP participate in cycles, any heat released from their oxidation and hydrolysis, respectively, is canceled out by their subsequent regenerative synthesis. If the authors are referring to energy that is used inefficiently, then that should be stated. Besides, it’s not always just heat – all life exports entropy, another form of energy.

We thank the reviewer. The text has been corrected and all reference to “heat” has been removed.

Line 55: Several regions have been referred to as the world’s largest biome: global marine sediments, oceanic basement, and the terrestrial subsurface, all of which are larger than an ocean gyre. Just say that it’s one of the largest.

The text has been revised.

Line 85: black carbon is not highly oxidized. The average nominal oxidation state of carbon in polyaromatic hydrocarbons is < 0 , more reduced than glucose, which is 0. Even if black carbon were pure graphite, it’s oxidation state would be 0.

We thank the reviewer for making this important point about the nominal oxidation state of black carbon. The text has been corrected.

Line 92 this is a confusing way to report the stoichiometry of organic matter since O typically stands for oxygen in this context. Drop the O and say that this is the stoichiometry of CHNP in organic carbon; Furthermore, this is a rather incomplete analysis of the stoichiometry of natural organic matter, which includes a fair amount of oxygen.

The “O” was added to designate organic carbon as opposed to total carbon. We note the possible confusion and also agree with the point regarding lack of direct oxygen measurements as discussed above.

Lines 142-144: recalcitrance of organic carbon is a function of the ecosystem; fullerenes, black carbon and kerogens can be consumed by microorganisms given the right conditions. After all, the radio carbon age of black carbon in seawater that is noted is 20,000 years old, not radio-carbon dead.

We agree with the reviewer that most organic carbon is “biodegradable.” The term recalcitrant is no longer used.

Line 163: define SEP; don’t make the reader sift through another paper to define an acronym

SEP is now defined.

Line 184: wrong reference

The reference to Platt and Subba Rao was the correct citation. They collected suspended particulate matter samples during a phytoplankton bloom in St. Margaret’s Bay, Nova Scotia in 1969. They measured organic matter and total energy content by bomb calorimetry. Their specific energy estimate for carbon fixed during photosynthesis varied between 58-75 Joules mg^{-1} C with an overall mean of 66 Joules mg^{-1} C. This value is similar to our reported mean value of ~57 Joules mg^{-1} OC measured at the 4,000 m reference depth.

It’s quite a stretch to use data from one spot in the ocean to extrapolate how fluxes of particulate organic carbon will change in response to climate change. This requires more data to substantiate.

Both reviewers made this comment. We have reworded this comment in the revised manuscript.

Table 1: the last four columns should say that these are the fluxes of these elements in the organic matter in SPM, if that’s what they are.

Yes, that is what they are and the Table description now includes source information.

Figure 1: put some space between the figures.

The caption is vague; display items should be able to stand alone with their captions (i.e., define the acronyms); These are flux profiles and mass... of sinking particulate matter as a function of depth at the ALOHA station...;

If you’re going to dump model parameters in the caption, you have to show the related equation somewhere and define what the parameters are.

Thanks for this constructive criticism. We have not separated the panels, but we have restructured the figure and reworded the caption for clarity. The parameters refer to fits of the so-called Martin et al. curve which is nearly universally used to describe the depth-dependent attrition of particle flux in the sea. This is now explicitly stated for a more general audience.

Figure 2 : This is not the “total energy content” It is, I think, the enthalpies of combustion of organic carbon in SPM (total energy would include gravitational, kinetic, nuclear, etc.); same comment as for Fig 1 regarding model parameters and acronyms;

Fig. 2b needs more of an explanation; flux’s don’t have ‘arbitrary units’; perhaps these are non-dimensionalized fluxes?

We apologize for any confusion in the presentation and meaning of Figure 2. The fluxes at the bottom were shown in “arbitrary units” because each parameter has different units. The summary graph was meant to show the relative attrition patterns for each parameter. We now use the term “Normalized Flux.”

Figure 3c: this caption needs to be more specific; “this panel is a schematic of...”

As noted above, these are not chemical reactions and not all of them release heat.

Figure 3c, which shows a schematic of the interpretation of our results, has been revised.

Reviewers' comments:

Reviewer #1 (Remarks to the Author):

I was asked to comment on the revisions made to this paper based upon my earlier review.

I am pleased with the responses the authors provided and changes I could see in the revised manuscript.

I have no further comments and urge publication at this time.

Reviewer #3 (Remarks to the Author):

The additional evaluation as proposed by Reviewer 2 and completed by the authors on the nominal oxidation state of carbon adds strength to deciphering the potential thermodynamic controls on OM degradation in the water column.

However, with reference to the calorimetry data as it is presented and as previously stated by reviewer 2, there is a very important difference between the heat produced during the combustion of the SPM and the actual energy consumed by (i.e., Gibbs energy) the microorganisms. In lines 57-60 the authors offer the distinction, however, they do not guide the reader on how these values should be cautiously interpreted. As suggested by the reviewer 2, the authors should have discussed directionality and the role of entropy on the overall energy balance (i.e., 2nd law of thermodynamics), which they did not address in the revised manuscript. The authors may benefit from reading Von Stockar and Liu (1999) wherein they will find a relevant discussion on the relative roles of enthalpy and entropy on aerobic heterotrophic metabolisms, which illustrates that aerobic metabolisms (i.e., catabolism + anabolism) are predicted to be enthalpy driven with little contribution from entropy. However, it is important to note that Von Stockar and Liu are discussing Gibbs energies of metabolism. While not directly comparable, the authors' approach using combustion calorimetry, may be a pseudo indicator for the directionality of the reaction.

As for converting the calorimetric data to fluxes, typically, for heterotrophic metabolisms, energy fluxes refer to Gibbs energies (e.g., Hoehler and Jørgensen, 2013) fluxes which are ideally calculated under non-standard state conditions. My fear is that these "energy fluxes" will be over interpreted by the readers.

While the particular application of combustion calorimetry to collected SPM samples to predict the maximum potential energy gain is new, from a bioenergetics perspective, the results are not particularly novel. However, I do not work on marine systems. To me, from a bioenergetics perspective the evaluation of the NOSC is more compelling than the calorimetry data.

Hoehler, T. M., & Jørgensen, B. B. (2013). Microbial life under extreme energy limitation. *Nature Reviews Microbiology*, 11(2), 83.

Von Stockar, U., & Liu, J. S. (1999). Does microbial life always feed on negative entropy? Thermodynamic analysis of microbial growth. *Biochimica et Biophysica Acta (BBA)-Bioenergetics*, 1412(3), 191-211.

Response to Reviewer Comments

We thank the Editor and external reviewer for the time and effort they have invested. We are especially appreciative of the comments and constructive criticisms of our work which, we believe, have greatly improved our presentation. Below (in bold font) we provide our responses and detail the major changes that have been made in our revised manuscript.

Reviewer #3 (Remarks to the Author):

The additional evaluation as proposed by Reviewer 2 and completed by the authors on the nominal oxidation state of carbon adds strength to deciphering the potential thermodynamic controls on OM degradation in the water column.

However, with reference to the calorimetry data as it is presented and as previously stated by reviewer 2, there is a very important difference between the heat produced during the combustion of the SPM and the actual energy consumed by (i.e., Gibbs energy) the microorganisms. In lines 57-60 the authors offer the distinction, however, they do not guide the reader on how these values should be cautiously interpreted. As suggested by the reviewer 2, the authors should have discussed directionality and the role of entropy on the overall energy balance (i.e., 2nd law of thermodynamics), which they did not address in the revised manuscript. The authors may benefit from reading Von Stockar and Liu (1999) wherein they will find a relevant discussion on the relative roles of enthalpy and entropy on aerobic heterotrophic metabolisms, which illustrates that aerobic metabolisms (i.e., catabolism + anabolism) are predicted to be enthalpy driven with little contribution from entropy. However, it is important to note that Von Stockar and Liu are discussing Gibbs energies of metabolism. While not directly comparable, the authors' approach using combustion calorimetry, may be a pseudo indicator for the directionality of the reaction.

As for converting the calorimetric data to fluxes, typically, for heterotrophic metabolisms, energy fluxes refer to Gibbs energies (e.g., Hoehler and Jørgensen, 2013) fluxes which are ideally calculated under non-standard state conditions. My fear is that these "energy fluxes" will be over interpreted by the readers.

While the particular application of combustion calorimetry to collected SPM samples to predict the maximum potential energy gain is new, from a bioenergetics perspective, the results are not particularly novel. However, I do not work on marine systems. To me, from a bioenergetics perspective the evaluation of the NOSC is more compelling than the calorimetry data.

Author Response:

We thank Reviewer #3 for these helpful comments regarding the differences between enthalpy of combustion of SPM, which we report in our manuscript, and Gibbs energy, and for providing us with two relevant references. To better constrain the interpretation of our field data, we compared the enthalpies of combustion and Gibbs energies of combustion for a total of 91 diverse organic compounds as well as CO₂ under standard conditions (n.b., for CO₂, the numbers are of course zero in both cases). The enthalpies and Gibbs energies of formation were available from the literature in all cases. We calculated enthalpies and Gibbs energies of combustion for the organic compounds using the enthalpies of formation of water and CO₂ (−285.8 and −393.5 kJ mol^{−1}, respectively) and the Gibbs energies of formation of water and CO₂ (−237.2 and −394.4 kJ mol^{−1}, respectively). For example, in the case of methane, the literature enthalpy of formation and Gibbs energy of formation are −74.8 and −50.8 kJ mol^{−1}, respectively. The combustion of methane can be written as

which can be broken down into the following reactions:

The overall enthalpy of combustion is therefore

$$74.8 - 393.5 - 2(285.8) = -890.3 \text{ kJ mol}^{-1}$$

and the overall Gibbs energy of combustion is therefore

$$50.8 - 394.4 - 2(237.2) = -818 \text{ kJ mol}^{-1}$$

The literature values in all cases were available as a check on the calculated enthalpies of combustion. The 91 organic compounds included methane, ethane, propane, butane, pentane, hexane, heptanes, octane, nonane, decane, undecane, dodecane, 2-methyl propane, 2-methyl butane, 2-methyl pentane, 2-methyl hexane, 2-methyl heptanes, 2,2-dimethyl propane, cyclopropane, cyclopentane, cyclohexane, ethane, propene, but-1-ene, trans-but-2-ene, cis-but-2-ene, buta-1,2-diene, buta-1,3-diene, phenylethene, ethyne, propyne, benzene, methylbenzene, ethylbenzene, propylbenzene, 1,2-dimethylbenzene, 1,3-dimethylbenzene, 1,4-dimethylbenzene, ethenylbenzene, methyl amine, dimethyl amine, trimethyl amine, 1-amino butane, methanol, ethanol, propan-1-ol, propan-2-ol, butan-1-ol, pentan-1-ol, hexan-1-ol, heptan-1-ol, octan-1-ol, ethan-1,2-diol, cyclohexanol, methoxymethane, ethoxyethane, methanol, ethanol, propanal, butanal, acetone, butanone, methanoic acid, ethanoic acid, propionic acid, glutamic acid, glycine, serine, CO₂, urea, cyclooctatetraene, nitrobenzene, phenol, pyridine, oxalic acid, benzoic acid, valine, phenylalanine, threonine,

leucine, isoleucine, lysine, proline, tyrosine, methionine, cysteine, aspartic acid, asparagine, glucose, sucrose, fructose, and lactose.

We found that nearly all of the values (89%) for the enthalpies and Gibbs energies of combustion agreed to within $\pm 10\%$ (Figure 1, below). The two outliers were cyclohexanol (ratio of Gibbs energy to enthalpy of 1.40) and oxalic acid (ratio of Gibbs energy to enthalpy of 0.53). This analysis is consistent with the comment of Reviewer #3 that aerobic metabolisms are enthalpy-driven with little contribution from entropy. This analysis is now included in the text along with the von Stockar and Liu 1999 citation.

Figure 1: Comparison of enthalpy of combustion vs. Gibbs energy of combustion for 91 diverse organic compounds. The solid line is the 1:1 relationship and the dashed lines represent $\pm 10\%$. The two circled compounds are: (1) cyclohexanol and (2) oxalic acid.

We then developed a multiple linear regression model to estimate the standard Gibbs energy of combustion of organic compounds from the elemental composition of each compound. We first used this method to estimate the Gibbs energies of combustion from the H/C, N/C, O/C and S/C ratios of the diverse 91 organic compounds and CO₂. The result of that analysis is shown in Figure 2 (which is now also included in the Supplementary Information section of our revised manuscript).

Figure 2: Gibbs energy of combustion for 91 diverse organic compounds compared to Gibbs energy of combustion calculated using a model based on the elemental stoichiometry of each compound. The straight line is the 1:1 line.

This stoichiometric-based multiple linear regression model could then be applied to the field collected SPM as a means to independently estimate the standard Gibbs energies of combustion, and to compare them to our enthalpy measurements. The estimated standard Gibbs energies of combustion of the SPM collected in our sediment traps based on the measured elemental stoichiometry are shown in Figure 3.

In the upper 175 m of the water column, standard Gibbs energy of combustion averaged -31 J/mgC, with no significant correlation with depth ($p>0.15$). However, at depths from 200 to 500 m, there was a highly significant correlation with depth ($p=0.004$), and at 500 m the standard Gibbs energy of combustion was only -21 J/mgC. This relatively low value, and the trend with depth, compare favorably to our measured enthalpies of combustion (i.e., 40.50 J/mg C at 200 m to 11.09 J/mg C at 500 m), and with the ecological interpretations that we present in our manuscript, namely that the value of the organic matter in the particles as a source of energy for aerobic metabolism declined steadily with depth below the euphotic zone. However, there are several important caveats to this line of reasoning that we have now included in the discussion. First, these are standard Gibbs energies of combustion. The in situ energies would depend on the activities of the substrates and products as well as on temperature and pressure. Second, even a reaction that is favored thermodynamically may not occur for kinetic reasons. The apparently recalcitrant nature of the particles in the traps at 500 m may therefore be due as much or more to kinetic constraints than to thermodynamic considerations. And finally, that the suitability of the organic matter for anabolic metabolism will clearly depend on factors unrelated to its use for catabolism. We have addressed these three issues in the revised

manuscript to avoid over-interpretation by readers, the major concern noted by Reviewer #3.

Another expressed concern was our use of the term “energy flux” whose meaning in our manuscript may have been misinterpreted by Reviewer #3. The main theme of our paper is the biological carbon pump, which seeks to understand the mechanisms and processes responsible for controlling the downward flux of carbon (termed “carbon flux” in the relevant literature) and associated elements in the ocean. By analogy to downward carbon and mass fluxes we used the term “energy flux” to denote the rate of potential energy transport in units $\text{J m}^{-2} \text{d}^{-1}$. We have now clarified this meaning in the revised manuscript.

Finally, we respectfully disagree with the Reviewer’s comment about the novelty of our results. Our data on the enthalpy of SPM are the first of their kind in the discipline of marine biogeochemistry. To quote Reviewer #1 during the first round of evaluation (I am not sure if all reviewers have access to all reviews): *“I found this paper to be very novel, of high importance and well presented. It is novel because of the concept of tracking energy density, not simply C, mass or other fluxes on sinking particles out of the surface ocean.”*